# Gender-Specific Risk Factors and Prevalence for Sarcopenia among Community-Dwelling Young-Old Adults

**DOI:** 10.3390/ijerph19127232

**Published:** 2022-06-13

**Authors:** Jongseok Hwang, Soonjee Park

**Affiliations:** 1Institute of Human Ecology, Yeungnam University, Gyeongsan 38541, Korea; sfcsfc44@naver.com; 2Department of Clothing and Fashion, Yeungnam University, Gyeongsan 38541, Korea

**Keywords:** sarcopenia, young-old adults, risk factors, prevalence, odd ratio

## Abstract

Sarcopenia in the elderly is a serious global public health problem. Numerous sarcopenia studies classified their subjects into a single group, but health conditions and body composition vary according to age. This study examined the prevalence of sarcopenia according to gender and assessed the gender-specific risk factors in young-old adults. In this study, 2697 participants in Korea aged from 65 to 74 years were analyzed from Korea National Health and Nutrition Examination Surveys. The prevalence of sarcopenia in males and females was 19.2% (CI 95%: 16.4–22.3) and 26.4% (23.7–29.4), respectively. The risk factors in men were age, body mass index (BMI), waist circumference (WC), skeletal muscle index (SMI), fasting glucose (FG), triglyceride, and systolic blood pressure (SBP). Their odd ratios were 1.447, 0.102, 1.494, 0.211, 0.877, 1.012, and 1.347. The risk factors in women were age, height, weight, BMI, WC, SMI, and fasting glucose with values of 1.489, 0.096, 0.079, 0.158, 0.042, and 1.071, respectively. The prevalence of sarcopenia was higher in females than in males. Overall, the clinical risk factors in males were age, height, BMI, WC, SMI, FG, triglyceride, and SBP. Age, height, weight, BMI, WC, SMI, and FG were the risk factors for women.

## 1. Introduction

Sarcopenia is defined as the age-related loss of skeletal muscle mass that decreases muscle strength, function, and quality of life [1]. Although the definitive sarcopenia mechanism is unclear, several studies suggested that changing hormones, immobility, age-related muscle changes, nutrition, and neurodegenerative changes are possible contributing factors [2]. The elderly over 65 years old are more susceptible to sarcopenia. Skeletal muscle loss begins at 35 years of age, occurring at 1–2% every year. The muscle loss increases to 3% per year after 65 years [3].

The proportion of the elderly in Asia is increasing rapidly. In particular, Korea is the fastest aging nation in the world. Approximately 16.5% of the population was older than 65 years in 2021 and is expected to increase to 39.8% of those in 2050 [4]. Diseases related to aging, such as sarcopenia, will have a greater impact in Korea and Asia than in other countries. 

On the other hand, most sarcopenia studies classified subjects into a single group [5,6,7,8], despite the health condition and body composition of the elderly differing according to their age. Thus, dividing the elderly population according to age is crucial to proper investigation of the characteristics of sarcopenia. The ages of older adults can be divided into three categories: “young-old”, “old”, and “oldest-old” [9]. The age of young-old ranges from 65 to 74 years; the old ranges from 75 to 84 years old, and the oldest-old are over 85 years of age [4,10,11].

This is the first study to examine the young-old population aged 65 to 74 years. Understating the features of young-old people with sarcopenia is essential compared to those of the other counterpart ages. Because sarcopenia is frequently unrecognized and shows no signs and symptoms until it is severe, knowledge of the key feature of risk factors associated with early detection and prevention is very important [12]. Early diagnosis of sarcopenia focuses on detecting symptomatic patients as early as possible. By doing so, they have the best chance of effective treatment. When sarcopenia treatment is delayed or missed, there is a lower chance of a good quality of life, greater problems related to treatment, and higher costs of care.

Furthermore, several epidemiological results comparing older men and women revealed a discrepancy regarding prevalence [13,14,15,16,17,18]. They showed that sex-specific differences in absolute muscle loss rates are greater in men than in women, which cannot be attributed solely to the greater initial muscle mass in men [16,19,20].

Despite the lower prevalence of sarcopenia in women, Batsis et al., reported that sarcopenia is associated with a higher risk of death in older women. This raises the question of the potential differential sex-specificity that contributes to sarcopenia, requiring an in-depth understanding of the mechanism [21]. Therefore, this study aimed (1) to identify the prevalence of sarcopenia in young older people according to gender and (2) to assess the gender-specific risk factors in young-old people aged between 65 to 74 years. The study has two hypotheses: (1) the specific incident rate of sarcopenia in the young-old would differ according to gender; (2) gender-specific risk factors exist in sarcopenic young-old adults.

## 2. Materials and Methods

### 2.1. Datasets and Sampling

The present study used data from the 4th and 5th Korea National Health and Nutrition Examination Surveys (KNHANES) database. The datasets were collected by household interviews and standardized physical examinations administered at mobile examination centers. Community dwelling young-old, from 65 to 74 years old, with sarcopenia measurements and health surveys, were selected for the research. A stratified, multistage, clustered probability sampling method was applied to the data, representing the noninstitutionalized Korean general population. The study design is a cross-sectional study. The KNHANES database was obtained by the Korean Centers for Disease Control and Prevention Center (KCDCPC). All participants in the present study signed an informed consent form.

The KNHANES study examined 37,573 healthy people from January 2008 to December 2011. The present study excluded 33,535 people who were not 65 to 74 years of age. Of the remaining 4228 participants, 1364 and 167 subjects who did not undergo a sarcopenia examination and health survey, respectively, were excluded. Finally, 2697 participants were included in this study (Figure 1). Table 1 lists the general characteristics of the study subjects.

### 2.2. Variables

The present research used the following variables: age, height (cm), weight (kg), body mass index (BMI), waist circumference (WC), skeletal muscle index (SMI), smoking status, drinking status, fasting glucose, triglyceride, total cholesterol, systolic blood pressure, and diastolic blood pressure. The WC was the measured circumference passing a midpoint between the bottom of the rib cage and the top of the lateral border of the iliac crest with full expiration. The blood test was performed after eight hours of fasting. A mercury sphygmomanometer was used to measure the systolic blood pressure and diastolic blood pressure in the sitting position after a 10-min rest in a chair. Cigarette smokers and alcohol drinkers were categorized as non-users, ex-users, or current users.

### 2.3. Criteria for Sarcopenia

Sarcopenia was designated by the International Classification of Disease by World Health Organization (WHO), and its code was ICD-10-CM (M62.84). The presence of sarcopenia was determined by measuring the appendicular skeletal muscle mass (ASM) by dual X-ray absorptiometry (DEXA) (QDR4500A; Hologic, Inc., Bedford, MA, USA). The skeletal muscle mass index (SMI) was calculated as ASM (kg)/BMI (kg/m^2^). The SMI for sarcopenia determination was <0.789 in males and <0.521 in females, according to the Foundation for the National Institutes of Health Sarcopenia Project in the United States [22]. The investigator determined sarcopenia based on the calculated SMI. The validity and reliability of DEXA are well-established [23,24,25].

### 2.4. Data Analysis

The descriptive data are presented as the mean ± standard deviation. Complex sampling analysis was performed, adapting the weights given by KNHANES. Statistical analyses were performed using SPSS 22.0 window version (IBM Corporation, Armonk, NY, USA). Independent *t*-tests and chi-square analyses were performed to compare the chemical parameters of the sarcopenia and non-sarcopenia participants. Multiple logistic regression was exploited to calculate the odds ratio of sarcopenia of each sex. *p*-values < 0.05 were considered significant. 

## 3. Results

### 3.1. Prevalence of Sarcopenia in Young-Old

The male and female prevalence of sarcopenia in the weighted value was 19.2% (CI 95%: 16.4–22.3) and 26.4% (23.7–29.4), respectively (Table 2). Females had a higher prevalence than males. 

### 3.2. Clinical Risk Factors in Male 

Age, height, BMI, WC, SMI, fasting glucose, triglyceride, and systolic blood pressure were statistically significant (*p* < 0.05). By contrast, the weight, smoking status, drinking status, total cholesterol, and diastolic blood pressure variables were non-significant (*p* > 0.05) (Table 3). 

### 3.3. Clinical Risk Factors in Female 

The statistically significant clinical variables were age, height, weight, BMI, WC, SMI, and fasting glucose (*p* < 0.05). The smoking status, drinking status, triglyceride, total cholesterol, systolic blood pressure, and diastolic blood pressure variables are not statistically significant (*p* > 0.05) (Table 3).

### 3.4. Multiple Logistic Regression for Sarcopenia in Men

Separate multiple logistic regression analyses were conducted according to sex. This is because individual factors affect males and females differently. In men, multiple logistic regression for sarcopenia was performed as the outcome variable, choosing age, height, BMI, WC, SMI, fasting glucose, triglyceride, and systolic blood pressure. The odds ratios in age, BMI, WC, SMI, fasting glucose, triglyceride, and systolic blood pressure were statistically significant (*p* < 0.05). Their respective values were 1.447 (0.181–1.170), 0.102 (0.017–0.519), 1.494 (1.195–1.869), 0.211 (0.199–0.223), 0.877 (0.849–0.906), 1.012 (1.005–1.019), and 1.347 (1.276–1.421). The odds ratio for height was not statistically significant (*p* > 0.05) (Table 4).

### 3.5. Multiple Logistic Regression for Sarcopenia in Woman

The odds ratio of age, height, weight, BMI, WC, SMI, and fasting glucose were statistically significant with respective values of 1.489 (0.242–9.076), 0.096 (0.012–0.729), 0.079 (0.012–0.30), 0.158 (0.123–0.203), 0.042 (0.036–0.048), 1.071 (1.050–1.093) (*p* < 0.05). The odds ratio for age was not statistically significant (*p* > 0.05) (Table 5).

## 4. Discussion

This study examined the prevalence and risk factors according to gender in young older people with sarcopenia in Korea. The prevalence of sarcopenia in males and females was 22.8 and 26.4, respectively. The prevalence of sarcopenia was higher in males than in females. This finding is in line with several studies [26,27]. Dam et al., screened 10,063 people and reported a 5.10% and 11.80% prevalence of sarcopenia in men and women, respectively [26] (Dam et al., 2014). Similarly, Hunt et al., investigated 1921 community-dwelling older Japanese with a mean age of 73.0 years. Their sarcopenia prevalence was 10.34% in males and 16.56% in females [27]. 

A possible underlying mechanism for the lower prevalence in men was that many exogenous and endogenous factors affect the prevalence of sarcopenia. In particular, hormone changes promoting skeletal muscle loss are faster in women than men. From 65 to 74 years, woman undergo a higher rate of diminishing sex hormones, such as estrogens and androgens, than men [28]. 

This finding is inconsistent with studies performed in the United States, Hong Kong, and Taiwan [29,30]. Brown et al., investigated U.S 4425 community-dwelling older people whose average age was 70.1 years. They reported that the prevalence of sarcopenia is 44.8% in men and 30.24% in women [29]. Similarly, Chan et al., assessed 3957 old Chinese people living in the community in Hong Kong. The incidence of sarcopenia in men and women was 9.30% and 5.30%, respectively [30]. 

Regarding the gender-specific clinical parameters related to sarcopenia, age is a risk factor for sarcopenia in both males and females. This result parallels numerous studies [2,3,14]. The possible theoretical rationale is that aging is associated with significant increases in the serum levels of inflammatory markers and related factors in both sexes. Ferrucci reported that aging is related to significant increases in the serum levels of the inflammatory markers [31]. A chronic, sterile low-grade inflammation that develops with advanced age, in the absence of an overt infection, is related to the concept of immunosenescence [32,33]. Although inflammation is a crucial immune response against harmful pathogens in acute cases, these helpful acute inflammatory responses pose a problem in the elderly. This impaired acute response increases the susceptibility to infection, resulting in tissue degeneration, such as muscle tissue [31,33].

Waist circumference is related to sarcopenia in both sexes. This result is in line with previous sarcopenia studies [29,34,35]. A study on 4425 older adults in a community-dwelling study revealed odds ratios of 1.39 (1.05–1.84) in males and 1.44 (1.04–2.00) in females (95% CI) [29] in the hazard ratio. Confrortin et al., investigated 601 older adults and reported an odds ratio of 17.90 (1.48–201.16) (95% CI) in the anthropometric indicators, including waist circumference, waist to height ratio, and body mass index in both sexes [34]. Sanada et al., assessed 1488 Japanese adults and reported that men and women with sarcopenia have a significantly different waist circumference than males [35]. The possible underlying reason for such differences between sarcopenia and normal older adults is that decreased muscle mass and increased fat mass are interdependent [36]. Age-related muscle loss causes functional muscle weakness and muscle endurance, which results in a low level of physical activity [37]. This decreased muscle mass and physical activity is related directly to diminished total energy expenditure and prompt weight gain, especially in the abdominal area [37]. By contrast, increased fat mass, such as visceral fat, might generate high volumes of pro-inflammatory cytokines related to macrophages [38]. C-reactive protein and interleukin 6 associated with fat have a negative effect on muscle mass. Thus, the loss of muscle mass is strongly associated with increasing fat mass [39].

Fasting glucose is strongly associated with sarcopenia. This finding is consistent with precious sarcopenia studies [40,41,42,43,44]. A sarcopenic community-dwelling elderly cohort study of 157 people proved that the sarcopenic group has a higher incidence of impaired fasting glucose than the non-sarcopenic group [41]. Similarly, Ozturk et al., investigated 147 sarcopenia patients with an average age of 70.3 years. They found that sarcopenic patients had problems regulating their blood glucose levels [40]. The possible theoretical rationale is that the skeletal muscle plays a principal role in postprandial glucose regulation. Skeletal muscle absorbs up to 80% of glucose through insulin-dependent glucose uptake after ingestion. Insulin-dependent and independent skeletal muscle glucose processing requires glucose transport from the circulation to the muscle, glucose passing through the extracellular matrix to the cell membrane, and translocation at the cell membrane, constitutively or in response to insulin or exercise. The glucose gradient promotes uptake through the catalyzed glucose transporter, and glucose transport is regulated by the intracellular glucose metabolism [45]. The loss of skeletal muscle glucose uptake is related to an abnormal carbohydrate metabolism, which affects the fast glucose level.

The strength of this study is the specific gender risk factors in young-old adults. Although most studies evaluated the risk factors and prevalence [5,6,7,8], they classified their subjects into a single group [5,6,7,8]. The present study investigated the young-old population, providing a key feature of the risk factor in people aged between 65 to 74 years old. On the other hand, the present study had three shortcomings that should be considered for future research. One of the main limitations was that although 2697 subjects in this study represent the whole population by statistical weight, the risk factor driven by the cross-sectional design would be strengthened by a longitudinal study or randomized case-control study. Such studies can confirm the risk factors in sarcopenia. Another limitation is that although the present results are meaningful, this study investigated only young-old adults according to sex. To understand the characteristics of the young-old population better, it would have been better to conduct research on the old-elderly population at the same time to improve the quality of the research. Lastly, this study did not consider people with sarcopenia obesity or those who were osteosarcopenic obese. If those two conditions had been assessed, it would have provided a better understanding of the waist circumference and fast glucose level. Future studies will investigate these conditions.

## 5. Conclusions

The present study is the first clinical evidence demonstrating the gender-specific prevalence and clinical risk factors related to sarcopenia in young-old adults. The results showed that the prevalence of sarcopenia was higher in women than men, and its weighted value was 26.4% (23.7–29.4) and 19.2% (CI 95%: 16.4–22.3), respectively. The clinical risk factors in men were age, height, body mass index, waist circumference, skeletal muscle index, fasting glucose, triglyceride, and systolic blood pressure. The clinical risk factors for females were age, height, weight, body mass index, waist circumference, skeletal muscle index, and fasting glucose. The finding of specific risk factors in young-old will be very helpful in the early detection and treatment of sarcopenia. In particular, these results will be helpful to primary care clinicians and health care professionals when considering making a referral for the diagnosis and treatment of sarcopenia. They can easily recognize the likelihood that the person may be sarcopenic by understanding gender-specific prevalence and risk factors.

## Figures and Tables

**Figure 1 ijerph-19-07232-f001:**
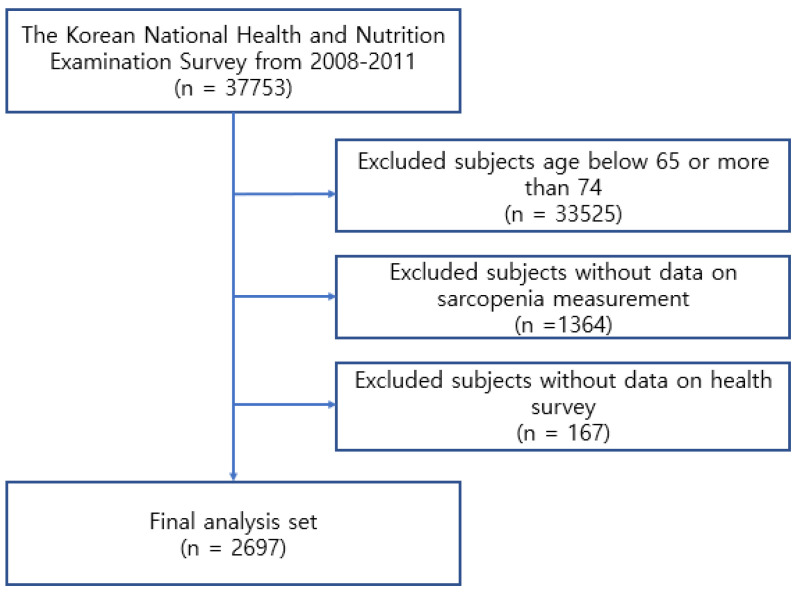
Flow chart for the selection of subjects.

**Table 1 ijerph-19-07232-t001:** Clinical characteristics of the study subjects (*n* = 2697).

Variables	Sarcopenia (*n* = 718)	Normal (*n* = 2146)	*p*
Gender (male/female) (%)	39.92/60.08	45.01/54.99	0.067
Age (years)	69.78 ± 2.704	69.19 ± 2.769	0.000 **
Height (cm)	153.05 ± 7.957	159.38 ± 8.447	0.000 **
Weight (kg)	60.20 ± 9.556	59.59 ± 9.841	0.000 **
Body mass index (kg/m^2^)	25.65 ± 3.228	23.39 ± 2.967	0.143
Waist circumference (cm)	87.72 ± 9.126	83.39 ± 8.936	0.000 **
Skeletal muscle index (kg/m^2^)	0.580 ± 0.128	0.735 ± 0.161	0.000 **
Smoking status (%) (current-/ex-/non-smoker)	25/13/62	29/13/58	0.048 *
Drinking status (%) (current-/ex-/non-smoker)	46/34/20	53/29/18	0.004 **
Fasting glucose (mg/dL)	107.90 ± 32.311	103.26 ± 24.465	0.000 **
Triglyceride (mg/dL)	159.73 ± 109.681	141.60 ± 88.007	0.000 **
Total cholesterol (mg/dL)	193.78 ± 38.744	189.81 ± 36.169	0.015 *
Systolic blood pressure (mmHg)	131.55 ± 17.604	130.01 ± 17.586	0.043 *
Diastolic blood pressure (mmHg)	77.35 ± 9.844	77.02 ± 9.830	0.429

The data are presented as the mean ± standard deviation. Independent *t*-test and chi square were significant at *p* < 0.05 *, *p* < 0.01 **.

**Table 2 ijerph-19-07232-t002:** Prevalence of gender-specific sarcopenia.

	Males			Females		
	Sarcopenia (*n* = 278)	Normal (*n* = 915)	Total	Sarcopenia (*n* = 401)	Normal (*n* = 1103)	Total
Un-weighted (%)	19.1	80.9	100	26.7	73.3	100
Weighted (%)	19.2 (16.4–22.3)	80.8 (76.8–82.7)	100	26.4 (23.7–29.4)	73.6 (70.6–76.3)	100

Weighed values present the 95% confidence interval.

**Table 3 ijerph-19-07232-t003:** Gender-specific clinical parameters related to sarcopenia.

	Males			Females		
	Sarcopenia (*n* = 278)	Normal (*n* = 915)	*p*	Sarcopenia (*n* = 401)	Normal (*n* = 1103)	*p*
Age (years)	69.700 ± 2.853	69.238 ± 2.794	0.014 *	69.826 ± 2.601	69.153 ± 2.750	0.000 **
Height (cm)	160.953 ± 4.986	166.942 ± 5.042	0.000 **	147.785 ± 4.424	153.197 ± 4.889	0.000 **
Weight (kg)	64.215 ± 9.266	64.148 ± 9.169	0.913	57.530 ± 8.789	55.850 ± 8.738	0.001 **
BMI (kg/m^2^)	24.717 ± 2.822	22.961 ± 2.706	0.000 **	26.270 ± 3.334	23.738 ± 3.123	0.000 **
WC (cm)	88.139 ± 8.855	84.352 ± 8.369	0.000 **	87.432 ± 9.302	82.611 ± 9.304	0.000 **
SMI (kg/m^2^)	0.731 ± 0.046	0.896 ± 0.073	0.000 **	0.480 ± 0.031	0.602 ± 0.061	0.000 **
Smoking status (%) (current-/ex-/non-smoker)	55.5/30.4/14.1	56.9/27.5/15.6	0.702	5.6/0.5/93.9	6.4/1.8/91.7	0.155
Drinking status (%) (current-/ex-/non-smoker)	71.9/19.7/8.5	72.1/16.8/11.2	0.368	31.1/20.6/48.3	37.2/19.0/43.7	0.157
FG (mg/dl)	109.726 ± 37.687	104.831 ± 26.780	0.016 *	106.638 ± 27.983	101.952 ± 22.291	0.001 **
Triglyceride	175.737 ± 143.986	136.534 ± 90.234	0.000 **	148.713 ± 76.028	145.801 ± 85.9360	0.548
TC	183.468 ± 36.724	179.928 ± 34.522	0.140	200.869 ± 38.547	197.992 ± 35.465	0.173
SBP (mmHg)	131.195 ± 17.868	128.717 ± 16.930	0.032	200.869 ± 38.547	131.065 ± 18.045	0.965
DBP (mmHg)	77.815 ± 10.159	76.953 ± 9.641	0.189	77.042 ± 9.628	77.066 ± 9.985	0.477

BMI: body mass index, WC: waist circumference, FG: fasting glucose, TC: total cholesterol, SBP: systolic blood pressure, DBP: diastolic blood pressure; The data is presented as the mean ± standard deviation; independent *t*-test and chi-square were significant at *p* < 0.05 *, *p* < 0.01 **.

**Table 4 ijerph-19-07232-t004:** Multiple logistic regression for sarcopenia in men.

Variables	Odd Ratio (95% of CI)	*p*
Age	1.447 (1.112–1.883)	0.006 **
Height	0.200 (0.033–1.224)	0.081
Body mass index (kg/m^2^)	0.102 (0.017–0.519)	0.031 *
Waist circumference	1.494 (1.195–1.869)	0.001 **
Skeletal muscle index	0.211 (0.199–0.223)	0.000 **
Fasting glucose	0.877 (0.849–0.906)	0.000 **
Triglyceride	1.012 (1.005–1.019)	0.000 **
Systolic blood pressure	1.347 (1.276–1.421)	0.000 **

The data is presented as the mean ± standard deviation. Multiple logistic regression was significant *p* < 0.05 *, *p* < 0.01 **.

**Table 5 ijerph-19-07232-t005:** Multiple logistic regression for sarcopenia in women.

Variables	Odd Ratio (95% of CI)	*p*
Age	1.398 (0.974–2.006)	0.069
Height	1.489 (0.242–9.076)	0.000 **
Weight	0.096 (0.012–0.729)	0.000 **
Body mass index	0.079 (0.012–0.30)	0.000 **
Waist circumference	0.158 (0.123–0.203)	0.000 **
Skeletal muscle index	0.042 (0.036–0.048)	0.000 **
Fasting glucose	1.071 (1.050–1.093)	0.000 **

The data is presented as the mean ± standard deviation. Multiple logistic regression was significant at *p* < 0.01 **.

## Data Availability

All data were anonymized and can be downloaded from the website at https://knhanes.kdca.go.kr/knhanes, accessed on 1 January 2022.

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
