# Peer review of "Gender-Specific Risk Factors and Prevalence for Sarcopenia among Community-Dwelling Young-Old Adults"

_ijerph, 2022, doi:10.3390/ijerph19127232_

Round 1
Reviewer 1 Report
1) Please divide the phenotype of sarcopenia in sarcopenia and sarcopenic obesity
2) into the discussion please cite and discuss this paper :
-Perna S, Peroni G, Faliva MA, Bartolo A, Naso M, Miccono A, Rondanelli M. Sarcopenia and sarcopenic obesity in comparison: prevalence, metabolic profile, and key differences. A cross-sectional study in Italian hospitalized elderly. Aging clinical and experimental research. 2017 Dec;29(6):1249-58.
3)table 1 : missing many units of measure - please fix it
4) sarcopenia and sarcopenic obesity must be divided in all tables based on smi and bmi criteria.
5) what about the osteosarcopenic obese people? did you have data about the bone mineral density? please discuss the following paper into the introduction since the sarcopenia could be a trigger for mortality
-Seo, J.A., Cho, H., Eun, C.R., Yoo, H.J., Kim, S.G., Choi, K.M., Baik, S.H., Choi, D.S., Park, M.H., Han, C. and Kim, N.H., 2012. Association between visceral obesity and sarcopenia and vitamin D deficiency in older Koreans: the Ansan Geriatric Study. Journal of the American Geriatrics Society, 60(4), pp.700-706.
6) Since you have defined the waist circumference , you must divide the phenotype of visceral sarcopenic obesity. Please define this phenotype in according to the following studies citing them into the methodology:
-Perna, S., Spadaccini, D., Nichetti, M., Avanzato, I., Faliva, M.A. and Rondanelli, M., 2018. Osteosarcopenic visceral obesity and osteosarcopenic subcutaneous obesity, two new phenotypes of sarcopenia: prevalence, metabolic profile, and risk factors. Journal of aging research, 2018.
-Jang, M., Park, H.W., Huh, J., Lee, J.H., Jeong, Y.K., Nah, Y.W., Park, J. and Kim, K.W., 2019. Predictive value of sarcopenia and visceral obesity for postoperative pancreatic fistula after pancreaticoduodenectomy analyzed on clinically acquired CT and MRI. European radiology, 29(5), pp.2417-2425.
Author Response
Dear editor,
At first, authors express their sincere gratitude to the reviewer’s valuable comments. Authors admire your deep and shard knowledge of sarcopenia and sarcopenia obesity. We were able to learn the way to write paper correctly because of your delicate comments with references. We really appreciate it with our whole hearts.
1)Please divide the phenotype of sarcopenia in sarcopenia and sarcopenic obesity
Author response: Thank you very much for your suggestion. reviewer’s comment and we would like to give the following explanation. We were not able to access low data of the Korean Centers for Disease Control and Prevention Center at this time. We have problem to assess analyses data Disease in the Control and Prevention Center because of COVID-19. We have to visit the center and analyze it. However, this problem would be fix soon. Authors promise to conduct research about sarcopenia obesity in the future.
2) into the discussion please cite and discuss this paper :
-Perna S, Peroni G, Faliva MA, Bartolo A, Naso M, Miccono A, Rondanelli M. Sarcopenia and sarcopenic obesity in comparison: prevalence, metabolic profile, and key differences. A cross-sectional study in Italian hospitalized elderly. Aging clinical and experimental research. 2017 Dec;29(6):1249-58.
Author response: Thank you so much for introducing the reference. We cited and added it in the discussion (lines 406-408, page 18 )(Please see the attached revised manuscript).
3) table 1 : missing many units of measure - please fix it
Author response: Authors feel sorry for missing unit of measure in the original manuscript. This was fixed (Table 1 in page 5, Table 3 in page 8, Please see the attached revised manuscript, the corrected part is changed to blue color).
4) sarcopenia and sarcopenic obesity must be divided in all tables based on smi and bmi criteria.
Author response: Authors totally agree with the reviewer’s valuable comments. We inputted those one in the limitation part of discussion (line 241-243, page 13). Though we were not able to access low data of the Korean Centers for Disease Control and Prevention Center at this time, authors promise to conduct research about sarcopenia obesity and bone mineral in the future.
5) what about the osteosarcopenic obese people? did you have data about the bone mineral density? please discuss the following paper into the introduction since the sarcopenia could be a trigger for mortality
Author response: Authors totally agree with the reviewer’s valuable comments. We inputted those one in the limitation part of discussion (line 241-243, page 13). The bone mineral data exist in the Korean Centers for Disease Control and Prevention Center. Though we were not able to access low data of the Korean Centers for Disease Control and Prevention Center at this time, authors promise to conduct research about sarcopenia obesity and bone mineral in the future.
6) Since you have defined the waist circumference , you must divide the phenotype of visceral sarcopenic obesity. Please define this phenotype in according to the following studies citing them into the methodology:
Author response: Authors fully agree with the reviewer’s valuable comments. However, we were not able to access low data of the Korean Centers for Disease Control and Prevention Center at this time. We inputted this point in the limitation part of discussion (line 241-243, page 13)
Thank you very much for the reviewer’s so many valuable comments. Thanks to the reviewer, the authors have improved their understanding of this research contents. Once again, thank you.

Reviewer 2 Report
The current study can further advance the knowledge on sarcopenia among young-old adults. The study has been well-designed, having a considerable simple size for the intended purpose. The results are consistent with other studies on the same issue, making the comparison feasible and informative.
Statistical analysis could be improved with the calculation of effect sizes.
Quality of the scientific and academic writing style is sufficient, even tough could be improved.
Be consistent with the use of males and females throughout the manuscript. Female in tables should be females.
Do not start sentence with numbers. Write numbers in words.
By the end of introduction, more practical implications from the findings of the study could be implemented.
Author Response
The current study can further advance the knowledge on sarcopenia among young-old adults. The study has been well-designed, having a considerable simple size for the intended purpose. The results are consistent with other studies on the same issue, making the comparison feasible and informative.
At first, authors express their deep gratitude to the reviewer’s valuable comments. We know it is arduous job for review papers. It consumes lots of time and effort. We were able to learn the way to write paper correctly because of your delicate comments. We really appreciate it with our whole hearts.
- Statistical analysis could be improved with the calculation of effect sizes.
Author response: Authors feel sorry for missing the calculation of effect size. We calculated effect size (Cohen's d) of Skeletal muscle index between sarcopenia(n=718) and normal group(n=2,146). This because skeletal muscle index is one of key indicator of sarcopenia. The analysis showed that Cohen's d = 1.065739. Which indicate it has large effect size (more than 0.8)
- Quality of the scientific and academic writing style is sufficient, even tough could be improved.
Author response: Authors fully agree with the editor’s comment. The attached revised manuscript has been proofread by native English academic advisor again.
- Be consistent with the use of males and females throughout the manuscript. Female in tables should be females.
Author response: Authors entirely agree with the reviewer’s comment. We corrected it (Table 2-3, pages 7-8 ).
- Do not start sentence with numbers. Write numbers in words.
Author response: Authors agree with the reviewer’s comment. We corrected it. (line 219, page 12).
5. By the end of introduction, more practical implications from the findings of the study could be implemented.
Author response: Authors entirely agree with the reviewer’s comment (lines 253-256, page 14).

Reviewer 3 Report
Thank you for giving me this opportunity to review this article. The article is well written, though I have some serious concerns regarding the article.
Mention the title more self-explanatory and clear.
Abstract:
- Provide the proper and adequate need of the study.
- Mention the type of study design.
- Mention the study duration and study setting.
- Include the character of study participants.
- Mention the statistical tests used for the study.
- Avoid abbreviation in the conclusion.
- The conclusion should be concise and self-explanatory and drawn on the basis of study reports.
- Mention the prevalence rate specifically in the conclusion part.
Manuscript
- The novelty of the study is missing, include more recent references emphasizing the need of this study.
- Include the clinical significance of this study over clinicians, patients, and researchers
- Mention the study hypothesis.
- Follow the strict author guidelines to present the paper. (STROBE)
- Mention the type of study design.
- Mention the study duration and study setting.
- Include the character of study participants.
- Include the criteria for diagnosing sarcopenia and who has diagnosed this condition?
- Mention the ICD code of Sarcopenia.
- Table 1: The p value for the gender row is not clear – what it indicates?
- Include the reliability and validity of the outcome measures used for this trial.
- Include the sample size calculation and the reference study for measuring the sample size.
- The results should be presented with 95%CI (upper limit – lower limit) for all the variables.
- Avoid abbreviations in the conclusion.
- The conclusion should be concise and self-explanatory and drawn on the basis of study reports.
- Mention the prevalence rate specifically in the conclusion part.
Add more recent references between 2017 – 2022.
Author Response
At first, authors express their deep gratitude to the reviewer’s valuable comments. And we are very appreciated for waiting the revision, after asking the editorial office to extend the deadline due to English proofreading.
0. Mention the title more self-explanatory and clear.
Authors response: It was amended (lines 5-6, page 2).
Abstract:
- Provide the proper and adequate need of the study.
- Mention the type of study design.
- Mention the study duration and study setting.
- Include the character of study participants.
- Mention the statistical tests used for the study.
- Avoid abbreviation in the conclusion.
- The conclusion should be concise and self-explanatory and drawn on the basis of study reports.
- Mention the prevalence rate specifically in the conclusion part.
Author response: Authors totally agree with the reviewer’s eight comments about abstract. However, according to the MDPI regulations, abstracts are limited to a maximum of 200 words.
Authors feel sorry that the editor's comments cannot be fully reflected due to lack of the space, although we reflected above things in the manuscript. (Please check the attached manuscript)
Manuscript
- The novelty of the study is missing, include more recent references emphasizing the need of this study.
Authors response: Thank you very much for so many valuable comments. Author amended it with recent reference (lines 38-50, page 2).
- Include the clinical significance of this study over clinicians, patients, and researchers
Authors response: It was amended (lines 253-256, page 14).
- Mention the study hypothesis.
Authors response: Authors feel sorry for missing hypothesis. We added the hypothesis (lines 59-61, page 3).
- Follow the strict author guidelines to present the paper. (STROBE)
Authors response: Authors entirely agree with the reviewer’s comment. We download STROBE guideline and followed it.
- Mention the type of study design.
Authors response: Authors feel sorry for missing the study design. The study design was cross-sessional design. We mentioned in the title (lines 5-6, page 2), and materials and methods (lines 71-72, page 3).
- Mention the study duration and study setting
Authors response: Authors feel sorry for confusing the reviewer.
Study duration was from January of 2008 to December of 2011 (line 75, page 3).
Study setting The datasets were collected by household interviews and standardized physical examinations administered at mobile examination centers (lines 67-71, page 3).
- Include the character of study participants.
Authors response: Authors feel sorry for confusing the reviewer. Target pogulation comprises non-institutionalized Korean community dwelling young old people aged between 65 to 74 years residing in Korea (lines 69-71, page 3).
- Include the criteria for diagnosing sarcopenia and who has diagnosed this condition?
Authors response: 1) Authors feel sorry for confusing the reviewer. In order to determine sarcopenia, skeletal muscle mass index is used. The skeletal muscle mass index (SMI) was calculated as appendicular skeletal muscle mass (ASM) (kg)/ body mass index(BMI) (kg/m2). The skeletal muscle mass index (SMI) < 0.789 in males and < 0.521 in females is used for screening sarcopenia, according to the Foundation for the National Institutes of Health Sarcopenia Project in the United States (Studenski, S.A.; Peters, K.W.; Alley, D.E.; Cawthon, P.M.; McLean, R.R.; Harris, T.B.; Ferrucci, L.; Guralnik, J.M.; Fragala, M.S.; Kenny, A.M.J.J.o.G.S.A.B.S.; et al. The FNIH sarcopenia project: rationale, study description, conference recommendations, and final estimates. 2014, 69, 547-558) (lines 112-114, page 6).
2) Authors add who determined sarcopenic condition. To be specific, first author (Jongseok Hwang) is health profession and spent more than 10 years in Geriatric Hospital in Korea. First author determine Sarcopenia based on calculated appendicular skeletal muscle mass data. And for data collection, professional medical investigator measured appendicular skeletal muscle mass (ASM) in the young old by dual X-ray absorptiometry (DEXA) (QDR4500A; Hologic, Inc., Bedford, MA) (lines 114-115, page 6).
- Mention the ICD code of Sarcopenia.
Authors response: Authors totally agree with the reviewer’s comment. We mentioned ICD-10-CM (M62.84) for sarcopenia ICD code (lines 109-110, page 6).
- Table 1: The p value for the gender row is not clear – what it indicates?
Authors response: This indicates that the male to female ratio is not statistically significant between the sarcopenia group and the normal group. In other words, the male to female ratio difference in the both groups are likely due to chance.
- Include the reliability and validity of the outcome measures used for this trial.
Authors response: Thank you. This was added (line 115, page 2).
The dual-energy X-ray absorptiometry (DEXA) is major outcome measure, screening sarcopenia. Especially, Glickman et al mentioned that DEXA also showed excellent reliability among three different operators to determine total, fat, and lean body mass (intraclass correlations, R = 0.94, 0.97, and 0.89, respectively)
These following articles demonstrate validity and reliability of DEXA.
- Kutáč, P.; Bunc, V.; Sigmund, M. Whole-body dual-energy X-ray absorptiometry demonstrates better reliability than segmental body composition analysis in college-aged students. PLoS One 2019, 14, e0215599-e0215599, doi:10.1371/journal.pone.0215599.
- Schubert, M.M.; Seay, R.F.; Spain, K.K.; Clarke, H.E.; Taylor, J.K.J.C.p.; imaging, f. Reliability and validity of various laboratory methods of body composition assessment in young adults. 2019, 39, 150-159.
- Dam, T.-T.; Peters, K.W.; Fragala, M.; Cawthon, P.M.; Harris, T.B.; McLean, R.; Shardell, M.; Alley, D.E.; Kenny, A.; Ferrucci, L.J.J.o.G.S.A.B.S.; et al. An evidence-based comparison of operational criteria for the presence of sarcopenia. 2014, 69, 584-590.
- Include the sample size calculation and the reference study for measuring the sample size.
Authors response: Authors feel sorry for confusing the reviewer. Authors have contemplated the reviewer’s comment (lines 70-72, page 3) and we would like to give the following explanation.
The target population of KNHANES comprises non-institutionalized Korean citizens residing in Korea. The sampling plan follows a multi-stage clustered probability design. For example, in the 2011 survey, 192 primary sampling units (PSUs) were drawn from approximately 200 000 geographically defined PSUs for the whole country (Ministry of Health and Welfare of Korea, Korea Centers for Disease Control and Prevention. 2011 Korea Health Statistics. Seoul: Ministry of Health and Welfare of Korea; 2012.). A PSU consisted of an average of 60 households, and 20 final target households were sampled for each PSU using systematic sampling; in the selected households, individuals aged 1 year and over were targeted. All statistics of this survey have been calculated using sample weights assigned to sample participants. The sample weights were constructed for sample participants to represent the Korean population by accounting for the complex survey design, survey non-response and post-stratification. The weights based on the inverse of selection probabilities and inverse of response rates were modified by adjusting them to the sex- and age-specific Korean populations (post-stratification).
- The results should be presented with 95%CI (upper limit – lower limit) for all the variables.
Authors response: Thank you very much for your suggestion. Authors have contemplated the reviewer’s comment and we would like to give the following explanation. ‘Table 1. Clinical characteristics of the study subjects’ present mean ± standard deviation format. And it deliver information sufficiently. Un-weighted prevalence of males and females in ‘Table 2. Prevalence of Gender-specific sarcopenia’ is just the number of sarcopenic people divided by all people. It does not have confidential interval. But weighted prevalence have the 95% confidential interval. This is because it calculated by complex sampling analysis via SPSS 22.
- Avoid abbreviations in the conclusion.
Authors response: This was corrected (lines 250, 252, page 14).
- The conclusion should be concise and self-explanatory and drawn on the basis of study reports.
Authors response: This was corrected (lines 247-256, pages13-14).
- Mention the prevalence rate specifically in the conclusion part.
Authors response: This was amended (line 249, page 14).
- Add more recent references between 2017 – 2022.
Authors response: We changed to recent reference as much as possible.

Round 2
Reviewer 3 Report
Dear authors,
I really appreciate all the authors for addressing all the comments in a very positive manner. The article is now in good condition to publish.
Regards